# Gut Microbiota Manipulation in Foals—Naturopathic Diarrhea Management, or Unsubstantiated Folly?

**DOI:** 10.3390/pathogens10091137

**Published:** 2021-09-04

**Authors:** Rachel Goodman-Davis, Marianna Figurska, Anna Cywinska

**Affiliations:** 1Faculty of Veterinary Medicine, Warsaw University of Life Sciences—SGGW, Nowoursynowska 166, 02-787 Warsaw, Poland; rachelgd@gmail.com (R.G.-D.); mariankafigurska@gmail.com (M.F.); 2The Scientific Society of Veterinary Medicine, Warsaw University of Life Sciences—SGGW, 02-787 Warsaw, Poland; 3Faculty of Biological and Veterinary Sciences, Nicolaus Copernicus University in Toruń, Lwowska 1, 87-100 Toruń, Poland

**Keywords:** foal, gut microbiota, diarrhea, probiotics, prebiotics

## Abstract

Diarrhea in foals is a problem of significant clinical and economic consequence, and there are good reasons to believe microbiota manipulation can play an important role in its management. However, given the dynamic development of the foal microbiota and its importance in health and disease, any prophylactic or therapeutic efforts to alter its composition should be evidence based. The few clinical trials of probiotic preparations conducted in foals to date show underwhelming evidence of efficacy and a demonstrated potential to aggravate rather than mitigate diarrhea. Furthermore, recent studies have affirmed that variable but universally inadequate quality control of probiotics enables inadvertent administration of toxin-producing or otherwise pathogenic bacterial strains, as well as strains bearing transferrable antimicrobial resistance genes. Consequently, it seems advisable to approach probiotic therapy in particular with caution for the time being. While prebiotics show initial promise, an even greater scarcity of clinical trials makes it impossible to weigh the pros and cons of their use. Advancing technology will surely continue to enable more detailed and accurate mapping of the equine adult and juvenile microbiota and potentially elucidate the complexities of causation in dysbiosis and disease. In the meantime, fecal microbiota transplantation may be an attractive therapeutic shortcut, allowing practitioners to reconstruct a healthy microbiota even without fully understanding its constitution.

## 1. Introduction

Diarrhea affects more than 50–60% of foals in the first six months of life [1,2] and bears significant health and economic consequences. Mortality rates vary widely with etiology, but foals are more susceptible to rapid and severe dehydration due to their small size and shorter colon, which is unable to reabsorb as much fluid from watery feces as that of an adult horse [3]. Common serious complications of disease processes producing diarrhea include sepsis and endotoxemia, which require intensive and expensive veterinary intervention and may even be fatal. Even mild diarrhea may have important economic and performance consequences due to reduced weight gain and poor growth. While diarrhea in foals is usually associated with enteritis and sepsis [1,4], not all diarrhea is pathological or warrants intervention. For example, one of the most prevalent types of diarrhea in young horses is foal-heat diarrhea [1,4], which is a non-infectious and usually mild occurrence seen in otherwise healthy foals and requires only close monitoring [4].

The gut microbiota plays an important role in disease processes and health maintenance in all mammalian species, but is of particular interest in horses [5,6,7,8,9,10]. As hindgut fermenters, horses are largely dependent on bacterial, protozoal and fungal activity in the cecum and colon for energy production and metabolic homeostasis [8]. Dysbiosis is associated with numerous pathologies both within and beyond the gastrointestinal tract (GIT), but differentiating cause from effect has so far proved difficult [6,7,10]. A 2020 Mach et al., study of the microbiota–gut–brain axis goes further and links variations in the microbiome of performance horses with stress, behavioral issues, performance and general welfare [8]. While this particular study focused on adult horses, it adds weight to a broader understanding of the importance of GIT microbiota in equine health and well-being.

In recent years, growing concern about antibiotic resistance in both veterinary and human medicine has shifted scientific and public attention towards “natural” or “alternative” therapies. Prophylactic and therapeutic manipulation of the gut microbiota is often extolled as a safe and low-cost way to manage intestinal disease. However, while probiotics, prebiotics and fecal microbiota transplants (FMT) have been extensively studied in humans, the body of research in horses is less expansive. What has been repeatedly demonstrated is that even microorganisms with proven cross-species efficacy cannot be presumed useful in horses [9,11,12]. Moreover, the foal microbiota undergoes rapid development from birth until approximately 60 days of age and continues to differ from that of mature horses for as long as 9 months [2,9,13,14]. Consequently, even results in adult horses cannot be extrapolated to foals.

Finally, poor regulation and quality control of microbial supplements [10,15,16], demonstration of mobile antibiotic resistance genes [17,18,19,20,21] and toxin production [18] in probiotic strains, as well as documented adverse effects in foals [9,22,23], call into question the perceived safety of such treatments. Given the importance of the poorly understood foal microbiota, it is imperative not to interfere with its development arbitrarily, as uninformed intervention may have unintended consequences. Thus, microbiota manipulation for prophylactic or therapeutic management of foal diarrhea should not be abandoned, but more research is needed for such therapy to be deemed either effective or safe.

## 2. Gut Microbiota Complexity in Horses

Over 30 years ago, Mackie et al., revealed the vastness of the equine microbiota, reporting ingesta in the cecum of adult horses contains approximately 10^9^ microorganisms per gram [24]. More recent technological advances, such as Next-Generation Sequencing, now permit culture-independent microbial identification; and advanced bioinformatics analysis and modeling have distinguished over 100 bacterial genera from seven phyla [8,25], as well as a myriad of protozoa, archaea and anaerobic fungi [8]. Yet, despite these advances, much remains unknown.

Numerous factors complicate attempts to map and manipulate the equine GIT microbiota. First, microbial populations and concentrations vary throughout the gastrointestinal tract [22,26,27], but for convenience reasons, studies generally rely on fecal samples, which reflect the microbial composition of only the right dorsal colon [22]. Microbial variation is also seen with age, body condition, sex, reproductive activity, breed, nutrition and management [5,6], general activity and stress level [8]; medications such as NSAIDs, antibiotics, antihelmintics and anesthetics [5,6]; and with exposure to foreign compounds like xenobiotics, polyketides and terpenoids found in the environment [28]. High individuality has even been reported among horses with similar diet and husbandry conditions [29]. Given the breadth of confounding factors, any assessment of “normal” microbial populations must be based on a large number of subjects. However, most studies to date have analyzed relatively small sample groups. Furthermore, while gut microbial variation is seen in many pathologic conditions such as colitis [30], colic [31], diarrhea, equine gastric ulcer syndrome (EGUS) and equine metabolic syndrome (EMS), as well as laminitis and grass sickness [2,5], limited understanding of what constitutes “normal” and “abnormal” [6] makes causal relationships difficult to establish. Finally, variations in study design and analysis make comparison among results problematic or even senseless [32].

## 3. Singularity of the Developing Foal Microbiota

As if the adult equine microbiota were not sufficiently complicated, the foal microbiota is even more so. While potential in utero vertical transmission of maternal microbiome constituents has recently been demonstrated [33,34,35], microbial colonization is largely believed to begin at birth via exposure to the dam’s vagina, perineum and skin [14]. Colonization begins with facultative anaerobes, followed by strict anaerobes such as *Bacteroidaceae* and *Lactobacillus*, which predominate by two weeks of age [36]. Several low-abundant genera essentially disappear after the first day of life [14], while other populations appear or become more populous. The heightened diversity and dynamics of the newborn foal microbiota are thought to be due to exposure via the mare and the environment to transient, non-colonizing organisms [15]; adaptation to continuous changes in diet [6], as well as to a period of coprophagic behavior [5]. Development of the foal’s microbiota continues throughout adolescence, but stabilizes around two months of age [2,6,13,14]. Marked changes in the microbiota have been noted after weaning and are generally attributed to both stress and the changing diet [5,6,28].

In a study of 11 foals on a single farm, Costa et al., found the microbiota of neonates, as in adults, was composed mainly of *Firmicutes*, but with several low-abundant genera unique to this age group [14]. From 2–30 days of age, foals had significantly lower microbial diversity than older animals, with a predominance of *Akkermansia*, while a higher abundance of Fibrobacteres were observed in foals after weaning. Stabilization of the intestinal microbiota was seen by day 60, but the variety, abundance and selection of bacterial species present in foal feces continued to differ from that of mature mares on the same farm even at nine months of age. In contrast, a similar size study by Lindenberg et al., reported higher diversity in the gut microbiota in younger age groups at days seven and 20 compared with day 50, when an increase in *Bacteroides* occurred [13]. De La Torre et al., found higher levels of *Enterobacteriaceae* in diarrheic feces of seven-day-old foals, which was attributed to coprophagic activity typical of that age, as well as greater levels of *Bifidobacteriaceae*, which are known to utilize lactose and milk oligosaccharides, reflecting seven-day-old foals’ continuing dependence on milk [28].

## 4. Causes and Consequences of Diarrhea in Foals

The most commonly reported etiologic agents of foal diarrheal disease (Table 1) include rotavirus, *Clostridium perfringens* types A and C, *Salmonella* spp., *Clostridium difficile*, *Cryptosporidia*, and *Lawsonia intracellularis* [1,35,36,37,38]. Less common causes include coronavirus, *Rhodococcus equi* and *Strongyloides westeri* [1]. Frederick et al., found foals less than one month of age were more likely to suffer from *Clostridium perfringens* and idiopathic diarrhea, while rotavirus, *Salmonella* spp. and parasites were more often implicated in older foals [37]. Recent evidence suggests co-infections with multiple agents are more prevalent than previously known and may produce more severe gastrointestinal disease [35].

However, by far the most common type of early juvenile diarrhea is foal-heat diarrhea, which is seen in 75–85% of otherwise systemically healthy foals between five and 15 days of age [1]. The loose feces were previously believed a response to compositional changes in the mare’s milk as she returned to a normal estrus cycle in the postpartum period. However, research has shown no such correlation, and a similar diarrheic phenomenon is seen in foals bottle-fed milk replacer [39,40]. Recent studies have revealed this time period to be one of dynamic changes in the foal’s developing microbiome as the flora shift from those aiding digestion of milk to those better suited to a diet of solid fiber and carbohydrates [5,11,28,41]. Thus, foal-heat diarrhea is currently understood to result from normal physiologic changes in the GIT as the foal begins to ingest new types of feed [1,40,42], including the dam’s feces. John et al., posit as foals begin to eat solid, fibrous feeds, the hemicellulose and cellulose in fiber could be responsible for higher water-binding capacity and reduction in intestinal passage, thereby altering intestinal absorptive and secretory processes, resulting in increased free water in the colon and softer feces [11]. Another possible explanation is that diarrhea at this time is due to osmotically-active metabolites of bacterial digestion of fiber [11]. Masri et al., suggest the mechanism might involve small intestinal hypersecretion of electrolytes for which the absorptive capacity of the immature colon is unable to compensate [43]. Researchers have suggested diarrhea at 2–4 weeks of age may even be a prolonged foal-heat diarrhea caused by enduring microbiota instability [34]. Because these foals are otherwise healthy and the diarrhea tends to be mild and self-limiting, generally no treatment is required. However, individual foals may suffer more severe and prolonged diarrheic episodes [1], so foals should be closely monitored for any deterioration, including interruption of normal nursing [40,41].

Regardless of the pathogenesis, significant compositional alterations and decreased diversity of fecal microbiota across all taxonomic levels have been reported in cases of acute diarrhea in both adult horses and foals [2]. Schoster et al., reported feces of diarrheic foals were poor in *Lachnospiraceae* and *Ruminococcaceae*, members of the Clostridia class involved in gastrointestinal health in many mammalian species [2]. However, causation between shifts in bacterial relative abundance and observed physiological changes has thus far been difficult to establish due to technological limitations, which have only recently begun to improve [6].

**Table 1 pathogens-10-01137-t001:** Pathomechanisms of common etiologic agents of foal diarrhea.

Etiologic Agent	Mechanism of Action
Rotavirus	Primarily malabsorptive due to infection of mature absorptive villous enterocytes of small intestine, but toxin-mediated secretory component also contributes [1]. The NSP4 non-structural glycoprotein may contribute to the development of diarrhea via non-competitive inhibition of the Na+-D-Glucose symporter, preventing water reabsorption even in the absence of histological damage to the villi [44].
*Clostridium perfringens* type A	α-Toxin is produced, but not thought to be a significant enteric virulence factor [44]. β_2_-Toxin causes hemorrhage and necrosis of the intestinal wall, but has not been reported as a pathogenic factor in foal diarrhea [44]. Enterotoxin is produced by 2–6% of all *C. perfringens* isolates and types, but its role as a virulence factor remains unclear [44].
*Clostridium perfringens* type C	α-Toxin is produced, but not thought to be a significant enteric virulence factor [44]. β-toxin [1] causes hemorrhage and necrosis of the intestinal wall [44]. Enterotoxin is produced by 2–6% of all *C. perfringens* isolates and types, but its role as a virulence factor remains unclear [44].
*Salmonella* spp.	Colonizes the intestine via the invasion-associated type III secretion system, producing massive mucosal damage of the ileum and colon [1]. Additional virulence factors include entero- and cytotoxins, which stimulate severe local and systemic inflammatory responses [44].
*Clostridium difficile*	Produces several hydrolytic enzymes and at least five toxins [44]. Enterotoxin TcdA and cytotoxin TcdB cause increased paracellular permeability of mucosal surfaces, cell rounding, and eventually cell death or apoptosis [1]. Toxins disrupt the enterocyte cytoskeleton and tight junctions and cause severe inflammation of the lamina propria, as well as micro-ulceration of the colonic mucosa [44]. Toxin-induced inflammation increases fluid exudation and mucosal damage, resulting in diarrhea or pseudomembranous colitis [1].
*Cryptosporidium parvum*	Invades the microvillus border of the small intestine, primarily the ileum [44].
*Lawsonia intracellularis*	Infects epithelial crypt cells, colonizing large areas of the intestinal epithelium. As a result, the normal villus structure is replaced by glandular epithelium composed of undifferentiated crypt cells with a poorly developed brush border [44].
Coronavirus	Targeted cytolytic destruction of the small intestinal villi and absorptive enterocytes results in death due to electrolyte disturbances [45].
*Rhodococcus equi*	Abdominal lesions include ulcerative enterocolitis and typhilitis over the area of the Peyer’s patches, granulomatous or suppurative inflammation of the mesenteric and/or colonic lymph nodes, or a single large abdominal abscess [46].
*Strongyloides westeri*	Cause catarrhal inflammation with edema and erosions of the small intestinal mucosal epithelium, resulting in malabsorptive diarrhea [47].
Sepsis	Diarrhea results from hemodynamic alterations leading to GI mucosal hypoperfusion, inflammatory mediators associated with SIRS, and dysmotility [1].

## 5. The Appeal of Microbiota Manipulation

The most important factors in proper foal management and prevention of diarrhea are sufficient and timely intake of high-quality colostrum and maintenance of hygienic environmental conditions [3,4], but prophylactic and therapeutic pharmaceutical management also play a significant role [48,49].

For many years, it was common practice to administer a three-day prophylactic course of antimicrobial drugs to neonatal foals [49]. However, Wohlfender et al., argue prophylactic antimicrobials are no longer needed due to improved management practices, and reported no difference in infectious disease incidence between treated and untreated foals [49]. Furthermore, countless studies in recent years have highlighted the dangers of prophylactic antibiotic use with respect to the perpetuation and exacerbation of widespread and multi-species antimicrobial resistance [50]. In the European Union, such concerns have culminated in the new Regulation (EU) 2019/6 on veterinary medicinal products [51], which will significantly curtail the use of antibiotics in animals, including horses, when it comes into effect in 2022.

Another response to concerns regarding antibiotic overuse has been interest in prophylactic probiotics and prebiotics. Probiotics are defined by the WHO and FAO as “Live microorganisms which when administered in adequate amounts confer a health benefit on the host” [52]. Several microorganisms, including yeast and bacteria, are used as probiotics in human and veterinary medicine [15,53,54]. In a similar vein, prebiotics are “substrate that is selectively utilized by host microorganisms conferring a health benefit” [53]. Based on the same principles, but taking a more comprehensive approach, FMT involves transfer of fecal suspension from a healthy donor to the bowel of a recipient [15], and successful applications in human medicine have understandably piqued interest in the veterinary community [29,54,55].

In addition to the desirability of reduced antimicrobial use, interest in prophylactic and therapeutic manipulation of gut microbiota has been driven by successful stabilization of favorable intestinal microbiota in other species [22] and a perceived high safety index [12]. The ease of oral pro- and prebiotic administration further adds to their appeal [56], and while FMT is a more involved procedure, it is still minimally invasive and requires only basic equipment. However, while adverse effects of probiotics are rare in all species, their impact on the microbiota of both adult horses and young foals is poorly understood, and negative outcomes have been reported in foals [9,22,23].

## 6. Equine Probiotic Strains

In order for probiotics to “confer a health benefit”, such as normalizing disrupted intestinal microbial communities and increasing beneficial bacterial populations [57], the administered strains must meet a number of criteria. First, sufficient numbers of probiotic organisms must survive the acidic gastric environment and resist bile digestion in order to reach the site of action [12]. Once in the intestines, organisms must be able to adhere to mucus and epithelial cells [12,15] or they will simply be excreted, conferring transitory if any effect. Four main mechanisms of beneficial probiotic action have been proposed [9,10,15,57]—modulation of the host innate and acquired immune system, antimicrobial production, competitive exclusion of pathogenic bacteria and inhibition or inactivation of bacterial toxins. Unfortunately, much of the understanding of probiotic mechanisms is based solely on in vitro studies, which cannot be extrapolated to in vivo effect without clinical trials [9,58,59].

It has been postulated that probiotic organisms should ideally be host species-specific in order to improve GIT survival and intestinal colonization [59], and it is generally presumed equine probiotics should target the cecum and colon as these are the main sites of microbial activity, fermentation and gastrointestinal disease processes in horses [10,15]. However, with respect to foal diarrhea, this assumption may not hold. For example, *S. westeri* helminths mature in the small intestine of foals, causing malabsorptive diarrhea [47]. Rotavirus, which commonly affects foals in the first weeks of life, targets the villi of the duodenum, jejunum and ileum [60], as does equine coronavirus [45]. Similarly, *Cryptosporidium parvum* affects the distal small intestine, especially the ileum [44]. *L. intracellularis* proliferative enteropathy likewise predominantly affects the small intestine [61]. A retrospective analysis of necropsies of eight foals and horses with a presumptive diagnosis of *C. perfringens* type C enterotoxemia showed gross lesions with variable distribution throughout the small intestine of seven of the subjects, while only four had lesions in the cecum and colon [62]. Thus, probiotics designed to support hindgut fermentation may not be appropriately targeted to the intestinal segments most affected by common etiologic agents of diarrhea in foals.

Even when probiotics target the proper site of action, studies in adult horses have not produced consistent results. On the one hand, yeasts targeting the hindgut have been shown to improve digestion and assimilation of feed, and perhaps as a consequence, to improve growth performance [50,59]. In the EU, several products containing the yeast *Saccharomyces cerevisiae* are approved for use in horses as “digestive enhancers” or “gut flora stabilisers” [6,22]. *S. cerevisiae* has been shown to improve colonic digestion of acid and neutral detergent fiber in horses [50,59]. However, a 2013 study of *S. cerevisiae* supplementation in a dozen geldings found no effect on either fermentation profiles or fiber digestion [63]. Similarly, one study of *Saccharomyces boulardii* supplementation in 14 horses with acute enterocolitis showed a significant reduction in severity and duration of intestinal disease during hospitalization despite a lack of colonization [64]. However, in a study of 21 horses with antimicrobial induced diarrhea, *S. boulardii* supplementation had no statistical impact on the improvement of clinical parameters, length of hospital stay, incidence of secondary complications or survival [65].

Unfortunately, the most commonly used probiotic bacterial genera—*Lactobacillus*, *Bifidobacterium* and *Enterococcus*—are small intestinal commensals, comprising less than 1% of the large intestinal microbiota of healthy adult horses [15]. These genera are more widely distributed throughout the GIT of foals less than two-months-old, but are still relatively uncommon compared with other gut bacteria [15]. In general, the recently identified core components of the foal microbiota [2,5,13,14,26,28] differ markedly from probiotic products currently on the market. Lactobacilli and *Bifidobacterium*, popular probiotics in human health, are not consistently associated with equine GIT health, and their usefulness in horses has been questioned [12,15]. Bacteria commonly used as probiotics in other species (e.g., *Lactobacillus*, *Enterococcus*, *Bacillus*, *Streptococcus* and *Bifidobacterium*) have been evaluated in horses [9,11,12,15] with underwhelming results. However, a recent study of non-authochtonous *Enterococcus faecium* AL41, which produces antimicrobial bacteriocin—enterocin M, showed sufficient colonization of the equine GIT, reduction in *Aeromonas* populations and increased activity of hydrolytic enzymes [66].

The search for effective equine probiotic strains thus continues. The most abundant phylum isolated from the equine GIT in all age groups is *Firmicutes*, including *Ruminococcaceae* and *Lachnospiraceae*, which are Clostridia associated with equine gut health [15], prompting the suggestion Clostridia might be a more appropriate subject for future equine GIT probiotic research [10,15]. Additionally, a recent study of four strains of *Weissela confusa* showed promising in vitro results, leading the authors to conclude further study of these strains is warranted [67]. However, cautious optimism is advised given that in vitro success with other probiotic strains has repeatedly failed to meet expectations in vivo [9,23].

## 7. The Evolution of Probiotic Studies in Foals

While studies in adult horses may offer some transposable takeaways, foals require separate study due to their rapidly developing and divergent microbiota, particularly during the first two months of life. So far, studies of foals have revealed conflicting and inconclusive results, among which comparison is difficult due to variations in study parameters and metrics (Table 2).

In 2003, Weese et al., evaluated the ability of *Lactobacillus rhamnosus* strain GG (LGG), one of the most studied human probiotics, to colonize the intestines of adult horses and foals without causing adverse effects [12]. LGG had already been shown to survive acid and bile digestion and colonize the GIT of humans and had been used successfully to treat several forms of human diarrhea, including pediatric rotavirus. Yet despite previously demonstrated cross-species efficacy, intestinal colonization by LGG was at best sporadic and poor irrespective of the dose, although no adverse effects were seen in any of the study subjects [12]. Greater persistence of intestinal colonization was seen in foals than in adults, leading to speculation that as a human origin microorganism, LGG is not well adapted to compete with the indigenous intestinal flora of adult horses, but is less challenged by the immature gastrointestinal microbiota of the foal. The authors thus cautioned against extrapolating results in neonates to older foals with more mature microbiota.

Weese et al., postulated the reason probiotics had so far not been shown beneficial in horses might be due to improper selection of probiotic organisms and sought to identify lactic acid bacteria native to the equine intestinal tract for probiotic use [68]. Lactobacilli are predominant indigenous bacteria isolated from feces of yearlings and foals [34]. These bacteria naturally colonize the stratified squamous epithelium of the non-glandular area of the equine stomach, and in vitro observations have shown indigenous lactobacilli can attach host-specifically to keratinized epithelial cells of the equine stomach [34]. *Lactobacillus pentosus* WE7 (WE7) was isolated from 89% of foals’ feces and exhibited superior in vitro growth, inhibition of *E. coli*, moderate inhibition of *Streptococcus zooepidemicus*, and *C. difficile* and mild inhibition of *C. perfringens* [68]. While no obvious difference in colonization by WE7 was observed between foals and mature horses in the clinical trial [68], no comparison was made of recovery from mature and foal feces due to differences in dosing. The authors concluded WE7 warranted a randomized, blinded, placebo-controlled study of efficacy for use in the prevention and treatment of equine enteric disease, which they subsequently conducted with a cohort of 153 foals [23]. Despite the promising in vitro results, clinical monitoring showed foals treated with WE7 were significantly more likely to develop signs of depression, anorexia and colic, as well as to require veterinary examination and treatment. The study thus raised concerns regarding the use of probiotics tested only in vitro.

Working from the same premise, a trial in 54 neonate Thoroughbreds examined the effect and safety of a host-specific probiotic preparation containing 5 strains of lactobacilli isolated from healthy horses (*L. salivarius* YIT 0479, *L. reuteri* YIT 0480, *L. crispatus* YIT 0481, *L. johnsonii* YIT 0482 and *L. equi* YIT 0483) and suspected to be in a close symbiotic relationship with the equine GIT [34]. This *Lactobacillus* preparation caused no clinical side effects and led to a significant decrease in the incidence of diarrhea at three weeks of age, as well as a significant increase in body weight at one month of age. None of the treated foals required medical attention or antibiotics at three to four weeks-old. No significant differences were seen in the number of individual bacterial species identified in the two groups of fecal samples, but a tendency for earlier *Lactobacillus* intestinal colonization was observed in supplemented foals. Supplementation thus led to earlier recovery from foal-heat diarrhea, probably by aiding the establishment of normal intestinal microflora. The authors concluded probiotics for dietary supplementation of foals should contain only normal equine microorganisms. However, the significance of these results has been questioned given that differences between the groups were significant only at one time point two to three weeks after probiotic administration had ceased and with no discernible clinical impact [15].

Similar success was replicated by Tanabe et al., with a multi-strain probiotic (LacFi) containing commensal lactobacilli and bifidobacteria isolated from Thoroughbred intestines and administered to 101 neonate Thoroughbred foals [69]. The formulation was found to have anti-inflammatory properties and to exhibit intestinal barrier protective activity. Clinically, LacFi decreased the incidence of diarrhea in the treatment group by 45% and effectively halved its duration. Benefits were observed at all ages, but were most pronounced around four weeks and 10–16 weeks of age [69]. However, the study was not blinded, the treatment and control groups differed markedly (*n* = 101 and *n* = 29, respectively), randomization and monitoring methods were not clearly described, and there was no quality control of the administered probiotic, obfuscating the actual concentration of viable organisms administered [15].

Believing the adverse effects of WE7 supplementation were related to overgrowth of lactobacilli in the poorly developed neonatal intestinal microbiota, researchers next sought to improve performance by choosing strains used in human probiotic formulations with in vitro evidence of *Clostridium* inhibition. The efficacy of *L. rhamnosus* (LHR 19 and SP1), *L. plantarum* (LPAL and BG112), and *Bifidobacterium animalis lactis* in the prevention of diarrhea and shedding of *C. difficile* and *perfringens* in foals was thus examined in a randomized, placebo-controlled field study [9]. No difference was observed in the incidence or duration of diarrhea between the two groups, but supplemented foals were again more likely to develop diarrhea requiring veterinary intervention, which the authors likewise attributed to potential overproduction of lactic acid. Despite in vitro inhibition of *C. difficile* and *C. perfringens*, no reduction in shedding was observed. However, the authors recognized a number of factors limiting interpretation of their results, including a relatively small number of farms sampled, an unexpectedly reduced number of participating foals, inconsistent recording of clinical signs at the time of diarrheic episodes and lack of recording of reasons for veterinary treatment throughout the study. A longitudinal continuation of this study using Next-Generation Sequencing techniques found only limited variation in relative abundance of families and species and no effect on alpha-diversity and community structure throughout the first six weeks of life, although the authors qualified this finding in light of the relatively small dose of administered probiotic in comparison to the horse’s complete intestinal microbiota [56]. Even in foals, phylogenetic analysis showed no increased abundance of probiotic-derived strains. LEfSe analysis, which emphasizes both statistical significance and biological relevance, revealed enriched numbers of *Lactobacillus* in the probiotic-treated foals at six weeks of age. However, while bacterial species diversity is thought to be an important factor in gastrointestinal health, no effect on diversity indices was observed. The authors concluded using *Lactobacillus*- and *Bifidobacterium*-based probiotics to prevent foal diarrhea could be futile, and suggested future research on equine probiotics might find more success with members of the Clostridia class or other species with greater abundance and significance in the equine microbiota.

Based on encouraging effects on intestinal health in other species, John et al., selected *Bacillus cereus* var. *toyoi* to reduce diarrhea by modifying the developing intestinal microbiota and mucosa in foals [11]. In contrast to results seen in other animals, 88% of foals developed diarrhea, and supplementation had no effect on bacterial microflora or hematologic parameters. However, all foals were systemically normal during diarrheic episodes, leading the authors to conclude the observed diarrhea was not infectious, but merely foal-heat. The significance of these results is also limited by the small sample size from a single farm. Furthermore, some foals were treated with antibiotics during the study, which may have affected fecal microbial profiles, and microbial isolation relied entirely on culture-dependent methods, which are liable to miss or over-represent microbial populations according to ease of culture. Interestingly, while Weese et al., suggested colonization by LGG may have been more successful in foals due to lack of inhibition by an immature microbiota [12], this study speculated *B. cereus* var. *toyoi* may have faced heavy competition in the hindgut thanks to non-selective entry of diverse microorganisms to the foal GIT [11].

Rivulgo et al., found some success supplementing 10 newborn foals with *Enterococcus faecalis* CECT7121. While 40% of the PG foals experienced mild to moderate diarrhea during the 12-day observation period, none of the TG foals developed diarrhea, and no adverse effects of supplementation were observed [70].

However, when Urubschurov et al., supplemented 16 foals with *Enterococcus faecium* and *Lactobacillus rhamnosus* throughout the first two weeks of life, treated foals again experienced increased frequency and duration of diarrhea, as well as slower growth [22]. Supplementation had no significant impact on fecal microbiota composition, but microbial profiles at the conclusion of the study were more similar in treated foals, suggesting probiotic administration may have resulted in selection of certain bacteria.

## 8. Safety of Probiotics

While probiotics are generally considered safe in both healthy and diseased adult horses, a number of concerns bear examination. First, a lack of regulation and quality control with respect to the concentration of viable organisms in commercial probiotic preparations has led to the discovery that some contain up to 100 times less active ingredient than is claimed on the label [10]. Consequently, even though doses up to three times the manufacturer’s recommendation have been reported safe [10], such a claim cannot be supported if the administered dose is in fact undetermined. Furthermore, the lack of quality control means probiotic preparations may inadvertently introduce additional and undesirable microbial species to the gut microbiota [72].

Another issue is the apparent inability to extrapolate results from in vitro to in vivo, from one species to another, or even from adult horses to foals. While there have been no published reports of enteric disease following probiotic administration in adult horses [10], and several published studies have demonstrated the safety of both commercially available and self-made probiotics in foals [12,22,23,70], adverse enteric effects have been reported in foals [9,14,20,49,71]. The impact of probiotics on the foal microbiota are not yet well enough understood to definitively explain such effects, but presumably the explanation lies in the significant differences between the enteric immune systems and microbiota of adult horses and foals, particularly during the first 30 days of life [15].

One of the drivers of interest in microbiota supplementation is a desire to reduce antimicrobial use in order to stem the spread of bacterial antimicrobial resistance. For example, probiotic GIT fortification is an attractive alternative to traditional prophylactic antibiotic administration. However, some bacterial strains used in probiotic preparations are now known to carry mobile antibiotic resistance genes, which can be transferred to host microbiota and later be excreted into the environment. Conjugative transfer of genes involving transposons and integrons is particularly worrisome, because they allow genetic shuffling both within the genome and between the genome and plasmids, which allows for cross-species transfer [21].

In some instances, probiotic antibiotic resistance genes can be an asset [20]. For example, intrinsic vancomycin and metronidazole resistance allow *Lactobacillus* to survive in antibiotic-treated hosts and thus be used to prevent antibiotic-associated diarrhea. Intrinsic resistance is not transferrable and so poses no particular risk to the host [19]. However, transfer of resistance-conferring genes has been observed in vitro and in laboratory animals both among *Lactobacillus* strains and between lactobacilli and other Gram (+) bacteria, including *Staphylococcus* [10]. The most common resistance genes observed code for tetracycline resistance, but resistance genes for chloramphenicol, macrolides, aminoglycosides and beta-lactams, all of which are commonly used in diarrheic foals, have also been reported [10,17]. Contradictory reports regarding the transferability of erythromycin resistance from lactobacilli demand further study and clarification [20]. However, several strains of the bifidobacteria species with European Food Safety Authority (EFSA) Qualified Presumption of Safety (QPS) status display antibiotic resistance phenotypes, which have, in many cases, been linked to specific antibiotic resistance genes [20]. The *tet*(W) gene for Tetracycline resistance is especially ubiquitous among strains of *B. longum* and *B. animalis* subsp. *lactis*. While the potential of bifidobacteria to transfer resistance genes to closely related bacteria has been demonstrated in vitro, no studies have tested the possibility of transfer to other enteric bacteria [20].

Enterococci, which are used to prevent and treat diarrhea in many animals, have been identified as common carriers of mobile genetic elements bearing multiple resistance genes [16], and additional concerns have been raised regarding their use as probiotics due to potential pathogenicity and a demonstrated ability to enhance adhesion of enteropathogens in vitro [68]. In 2012, Gouriet et al., raised concerns regarding the role of *L. rhamnosus* in human bacteremia [73]. Similarly, a 2020 study found nearly half of the 65 *Bacillus* spp. strains isolated from commercially available probiotics in China were capable of producing hazardous toxins in addition to containing multiple antimicrobial resistance genes coupled with mobile genetic elements [18]. Sixty percent of the Chinese *Bacillus* isolates showed hemolytic activity, and almost half were able to produce enterotoxins and various cytotoxic surfactin-like toxins, suggesting such strains could actually cause or aggravate diarrhea [18]. In vivo tests of these strains in mice produced sepsis, intestinal inflammation and liver damage.

In the EU, all probiotics are evaluated for resistance genes before receiving QPS status. However, QPS status is awarded to a bacterial species, and genomic content varies widely within species, including lactobacilli [17]. The risks posed to One Health by probiotic-mediated antibiotic resistance have thus led some researchers to call for a unified global effort to improve probiotic screening [18].

## 9. Prebiotic Use in Horses and Foals

The body of research supporting prebiotic use in horses and foals is even less developed than that supporting probiotics, and while some initial results are promising, others raise concerns. Recently, interest has grown regarding the influence of colostral oligosaccharides, which are also present in milk [74]. Experiments in human infants and animals have demonstrated prebiotic effects of galacto-oligosaccharides (GOS), including enhanced defensive immune responses and reduced incidence of infection. However, other studies have reported only temporary or non-significant effects [74,75,76,77].

Vendrig et al., examined Pattern Recognition Receptor (PRR) agonist activity of potential prebiotic oligosaccharide compounds [78]. Neonate foals have limited innate and adaptive immune responses, as well as immature gastrointestinal epithelial barrier function, which makes them vulnerable to disturbances of mucosal homeostasis during early intestinal microbial colonization. Excessive inflammatory responses and bacterial translocation into the bloodstream result in septicemia, which is the leading cause of death in neonate foals. PRRs recognize bacteria and downregulate cytokine release, contributing to mucosal homeostasis and enhanced epithelial barrier function. Evidence suggests selective PRR agonists likewise aid in the orchestration of foal gut colonization by limiting inflammatory responses and improving epithelial barrier function [78].

In human infants and laboratory animals, dietary supplementation with GOS has shown prebiotic action and long-term immunomodulation of both defensive and allergic immune responses [74]. Vendrig et al., conducted an in vivo pilot study of the effects of GOS in horses in which six pony foals received a commercially available GOS oral supplement throughout the first four weeks of life [74]. Monitoring was continued for an additional ten weeks. At day 28, peripheral blood mononuclear cells (PBMCs) derived from both groups of foals were challenged with lipopolysaccharide. The PBMCs derived from the treatment group showed significantly lower relative mRNA expression of pro-inflammatory cytokines IFN-gamma and IL-6. No undesirable effects of the GOS regimen were detected, but more clinical trials are needed to confirm and apply the attenuating effects of GOS treatment on equine pro-inflammatory immune responses [74].

Similarly, fructo-oligosaccharides (FOS) are commonly administered to adult horses in order to reduce the risk of hindgut dysbiosis [79]. Oral supplementation of prebiotic preparations containing FOS or mannan-oligosaccharides have improved digestibility of dry matter, crude protein, and non-digestible fiber [32]. Furthermore, FOS reduced colonic dysbiosis in adult horses after an abrupt change in diet and altered fecal volatile fatty acid concentrations toward propionate and butyrate [32]. However, the degradation of FOS to butyric acid and other short chain fatty acids begins in the stomach, potentially increasing gastric concentrations of butyric acid. Cehak et al., tested the effects of comparable concentrations of butyric acid in the equine stomach in vitro and found histopathomorphological changes in the glandular mucosa, as well as impairment of functional mucosal integrity in the squamous and glandular mucosa, and thus cautioned against the use of prophylactic FOS, particularly in horses at risk for EGUS [79].

While it seems there are few reports of adverse effects of prebiotic use, a paucity of in vivo prebiotic trials in horses means such risks cannot be conclusively ruled out. Transferrable antibiotic resistance is likewise less studied in prebiotics than in probiotics, but a recent study examining expanded-spectrum cephalosporin resistance in commensal *E. coli* from healthy horses in France found a significant distribution of multi-drug resistant IncHI1 plasmids carrying ESBL genes. The authors suggested these troubling findings might be mediated by short-chain-fructo-oligosaccharide prebiotic use in horses [80].

## 10. Fecal Microbiota Transplantation

Probiotic formulations are composed of at most a few strains of microorganisms constituting only a tiny fraction of the intestinal microbiota, which may limit their ability to influence the entire GIT. Fecal microbial transplants, on the other hand, comprise thousands of species representing a comprehensive healthy microbiome [29].

FMT has been successful in treating recurrent *C. difficile* infection (CDI) and ulcerative colitis in man [54,81], and anecdotal reports suggest promise for treatment of adult horses with acute colitis, chronic diarrhea due to post-antimicrobial CDI, or inflammatory bowel disease [10]. Proposed mechanisms for FMT efficacy in treatment of CDI are similar to those of probiotics, including competition for limited resources, direct elimination of *C. difficile*, neutralization of toxins and induction of immune-mediated resistance, as well as restoration of secondary bile acid metabolism in the colon [29]. A small 2014 study of horses with antibiotic-induced or undifferentiated colitis reported improved fecal consistency following FMT in three out of four recipients [29]. These results are supported by a larger and more recent study by McKinney et al., which assessed the clinical and microbiota responses to FMT in 12 hospitalized adult horses with colitis [55]. Prior to treatment, researchers found lower alpha-diversity and higher beta-diversity of the fecal microbiota of horses with colitis. FMT recipients saw a greater overall reduction in diarrhea, greater day-to-day improvement of diarrhea, and greater microbiota normalization at the conclusion of the study compared with untreated horses. A small 2020 study, also by McKinney et al., found FMT was associated with improvement of diarrhea, increased abundance of desirable Verrucomicrobia, and increased alpha-diversity of fecal microbiota in three diarrheic geriatric horses [82].

No studies have evaluated the use of FMT in diarrheic foals, but restoration of normal commensals could potentially improve IL-1β recruitment of neutrophils to the intestinal mucosa and protect against sepsis [29]. Given that young foals obtain their initial microbial population via contact with the dam and later via selective coprophagy of her fresh feces, the dam would make a suitable donor for FMT. No adverse effects of FMT have been reported in horses [29], however, specimens should be screened for viruses, as well as pathogenic and antibiotic resistant bacteria before introduction to the host [10].

## 11. Conclusions

Encouraging results and applications of gut microbiota manipulation in other species have generated considerable interest in the use of such methods to confer enteric protection and manage diarrhea in foals. However, both GIT function and gut microbiota composition differ markedly in horses from those of the most studied subjects, which likely explains why cross-species efficacy has proved frustratingly elusive in equine studies. In fact, due to the prolonged colonization and dynamic development of the highly variable equine microbiota, even results of studies in adult horses cannot be presumed valid in foals. Most notably, while in adult horses probiotics are generally considered safe albeit not particularly effective, clinical trials in foals show not only underwhelming efficacy, but also a demonstrated potential to aggravate rather than mitigate diarrhea. Furthermore, recent studies indicate insufficient quality control of commercial probiotics may enable inadvertent administration of toxin-producing or otherwise pathogenic bacterial strains, as well as strains bearing transferrable antimicrobial resistance genes.

Consequently, it seems advisable to approach probiotic therapy in particular with caution for the time being. Similarly, while prebiotics show initial promise with few reported adverse effects, an even greater scarcity of species-specific clinical trials makes it impossible to weigh the pros and cons of their use. Thus, in order to improve the efficacy and safety of probiotics and prebiotics in both adult horses and foals, more studies are needed in larger populations of horses of various ages. Researchers should be mindful of the need to conduct quality control of any commercial microbial preparation employed in such a study in order to verify the actual dose of active ingredients administered. For improved clarity, effort should also be made to differentiate among infectious and non-infectious, pathogenic and physiologic diarrhea. Finally, while advancing technology will surely continue to enable more detailed and accurate mapping of the equine adult and juvenile microbiota, FMT may be an attractive therapeutic shortcut in the interim, allowing practitioners to reconstruct a healthy microbiota even without fully understanding its constitution.

## Figures and Tables

**Table 2 pathogens-10-01137-t002:** Results of probiotic clinical trials in foals.

Results of Probiotic Clinical Trials in Foals								
Probiotic Strain/Formulation, Reference	Study Size	Dose	Frequency and Duration of Administration	Duration of Monitoring	Aim	Outcome	Adverse Effects	Study Limitations
*Lactobacillus rhamnosus* strain GG (LGG) [12]	3 groups of 7 adult horses 2 groups of 7 foals, 1–3 days of age	Group 1 adults = 1 × 10^9^ cfu/50 kg BW Group 2 adults = 1 × 10^10^ cfu/50 kg BW Group 3 adults = 5 × 10^10^ cfu/50 kg BW Group 1 foals = 2 × 10^10^ cfu/50 kg BW Group 2 foals = 1 × 10^11^ cfu/50 kg BW	SID for 5 days	15 days	To evaluate whether LGG can colonize the intestines of adult horses and foals.	Poor colonization in both adults and foals, but better in foals. No dose dependence.	None	No control group
*Lactobacillus pentosus* WE7 [68]	9 foals, 2 days of age 8 adult horses	Foals = 1 × 10^11^ cfu Adult horses = 7 × 10^11^ cfu	SID for 5 days	13 days	To identify lactic acid bacteria of equine origin with predetermined properties that might make them useful as therapeutic probiotics.	In vitro testing, WE7 performed thebest with respect to the ability to grow in the presence of oxygen, aid and bile, The isolate was inhibitory against *Salmonella* and *E. coli*, moderately inhibitory against *S. zooepidemicus* and *C. difficile* and midly inhibitory against *C. perfringens.* After demonstrating the ability to survive GIT passage, WE7 was thus selected for further study of its potential to prevent and treat enteric diseases in horses.	None	No control group
*Lactobacillus pentosus* WE7 [23]	TG = 70 foals PG = 83 foals. Both groups 1–2 days of age	2 × 10^11^ cfu	SID for 7 days	14 days	To evaluate efficacy of *L. Pentosus* WE7 for prevention of neonatal diarrhea in foals.	Did not prevent diarrhea.	TG foals significantly more likely to develop depression, anorexia and colic, as well as to require veterinary treatment. TG foals had more, but not significantly more days of diarrhea. Probiotic significantly associated with development of diarrhea as well as additional clinical abnormalities.	
*Lactobacillus salivarius* YIT 0479, *L. reuteri* YIT 0480, *L. crispatus* YIT 0481, *L. johnsonii* YIT 0482 and *L. equi* YIT 0483 [34]	TG = 27 foals PG = 27 foals Both groups 1–7 days of age	5 g of preparation containing 1–4 × 10^10^ viable bacteria	SID for 7 days	4 weeks	To examine effect and safety of a host-specific probiotic preparation.	Significantly increased body weight in foals 14–30 days of age. Significantly lower incidence of diarrhea in TG. No significant difference in number of individual bacterial species up to 2 weeks of age. Tendency for earlier colonization of Lactobacillus in TG. Significantly greater concentration of total SCFAs in TG only at 7 days of age. No significant effects on hematologic or biochemical parameters.	None	Significant differences seen only at one time point 2–3 weeks after probiotic administration had ceased.
*Lactobacillus reuteri* KK18, *L. ruminis* KK14, *L. equi* KK15, *L. johnsonii* KK21, *Bifidobacterium Boum* HU (LacFi) [69]	TG = 101 PG = 29 Both groups newborns to 20 weeks of age	Mixture of 8.6 × 10^9^ cfu	SID on days 2–5 of life. Then once a week for up to 4 weeks	20 weeks	To evaluate effect on diarrhea prevention.	Showed anti-inflammatory properties and intestinal barrier protective activity. Decreased incidence of diarrhea by 45% and halved duration.	None	Not blinded. Significant difference between size of TG and PG. Randomization and monitoring methods not clear. No quality control of probiotic.
*Lactobacillus rhamnosus* (LHR 19 and SP1), *L. plantarum* (LPAL and BG112), *Bifidobacterium animalis lactis* [9,56]	TG = 36 foals PG = 36 foals Both groups 3 days of age	1 × 10^9^ cfu of each strain of *L. rhamnosus* (LHR19 and SP1) and *L. plantarum* (LPAL and BG112) and 1 × 10^10^ cfu of *B. animalis lactis*	SID for 3 weeks	4 weeks	To evaluate effect on incidence of foal diarrhea.	No difference in incidence or duration of diarrhea. No difference in prevalence of *C. perfringens* shedding. No increased abundance of probiotic derived strains. Enriched *Lactobacillus* at 6 weeks old. No effect on diversity.	TG foals more likely to develop diarrhea requiring veterinary intervention.	Small number of farms sampled. Reduced number of participating foals. Inconsistent recording of clinical signs at times of diarrheic episodes. Lack of recording of reasons for veterinary treatment during study. Small dose of probiotic relative to complete microbiome.
*Bacillus cereus* var. *toyoi* [11]	High dose TG = 10 foals Low dose TG = 7 foals PG = 8 foals All groups 1 day of age	High dose = 2 × 10^9^ cfu Low dose = 5 × 10^8^ cfu	SID for 58 days	58 days	To evaluate whether supplementation would modify the developing intestinal microflora and consequently reduce diarrhea in foals.	No effect on occurrence of diarrhea and health status of foals.		Small sample size from single farm. Some foals received antibiotic treatment during the study period. Used culture-based methods for pathogen screening.
*Enterococcus faecalis* CECT7121 [70]	TG = 10 PG = 10 Both groups 1–3 days of age	1 × 10^10^ cfu/mL	SID for 6 days	12 days	To assess efficacy of supplementation with *E. faecalis* CECT7121 for prevention of neonatal diarrhea	No incidence of diarrhea in TG vs. mild to moderate diarrhea in 40% of PG.	None	
*Enterococcus faecium* and *Lactobacillus rhamnosus* [22,71]	TG = 18 PG = 16 Both groups 1 day of age	1.05 × 10^9^ cfu *E. faecium * and 4.50 × 10^8^ cfu *L. rhamnosus*	SID for 14 days	56 days	to investigate whether supplementation would influence the bacterial composition of faecal microbiota of foals. To investigate whether supplementation would prevent or mitigate foal-heat diarrhea.	No significant impact on composition of faecal microbiota. Prevented reduction in bacterial similarity between 2 and 8 weeks of age. No reduction in diarrhea within the first two weeks of life.	TG foals suffered more frequent and longer episodes of diarrhea than PG group.	

cfu—colony forming units; SID—single in day; TG—treatment group; PG—placebo group; BW—bodyweight; SCFA—short-chain fatty acids.

## Data Availability

This paper contains all data.

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
