# Peer review of "Gut Microbiota Manipulation in Foals—Naturopathic Diarrhea Management, or Unsubstantiated Folly?"

_pathogens, 2021, doi:10.3390/pathogens10091137_

Round 1

Reviewer 1 Report

This review by Goodman-Davis et al. summarized the published literature about gut microbiota manipulations for naturopathic diarrhea management. This is a very well-written manuscript and deserves a place in the special issue "The Equine Gastrointestinal Microbiome and Enterophatogens." Overall they discussed the literature well. However, I have some suggestions that I think will add some more features to this review. 

  1. In the causes and consequences of Diarrhea in the Foals section, the authors can add more pathological features about Diarrhea. For example, what are the molecular characteristics of that Diarrhea? Is it any different from humans?
  2. Points 4 (Gut microbiota co plenty in horses) and 5 (Singularity of the developing Foal microbiota) may be put upfront before points 2 and 3.
  3. Another big discussion point is how microbiota manipulation can lead to naturopathic diarrhea management mechanistically. From this review, if I cleared that further studies are needed to answer this question. However, the authors can get ideas from human and mice studies and propose a potential mechanism to work in foals.
  4. The authors can draw some figures on which microbiota is involved in diarrhea development. Sometimes, a figure can be helpful to understand significant points. 

Author Response

REVIEWER 1

Comments and Suggestions for Authors

This review by Goodman-Davis et al. summarized the published literature about gut microbiota manipulations for naturopathic diarrhea management. This is a very well-written manuscript and deserves a place in the special issue "The Equine Gastrointestinal Microbiome and Enterophatogens." Overall they discussed the literature well. However, I have some suggestions that I think will add some more features to this review. 

  1. In the causes and consequences of Diarrhea in the Foals section, the authors can add more pathological features about Diarrhea. For example, what are the molecular characteristics of that Diarrhea? Is it any different from humans?

We have added more pathological features, including the anatomical sites involved in the problem in the lines 289-300. We have also added a table (in section 4) with the mechanisms of diarrhoeas caused by the most common pathogens

  1. Points 4 (Gut microbiota co plenty in horses) and 5 (Singularity of the developing Foal microbiota) may be put upfront before points 2 and 3.

The order of sections has been changed accordingly

  1. Another big discussion point is how microbiota manipulation can lead to naturopathic diarrhea management mechanistically. From this review, if I cleared that further studies are needed to answer this question. However, the authors can get ideas from human and mice studies and propose a potential mechanism to work in foals.

We tried to avoid the discussion based on human and rodent studies due to the physiological differences. Although humans and rodent studies are much more advanced, we would like to avoid speculation. The article is long and we are affraid the another section will make it too long. However, we can add such section if the Reviewer insists.

  1. The authors can draw some figures on which microbiota is involved in diarrhea development. Sometimes, a figure can be helpful to understand significant points. 

We have added another table (table 1, section 4) with pathomechanisms of diarrhoea caused by the most common pathogens. Due to the fact that this is still evolving knowledge and there isn’t enough of a well understood and universally accepted consensus we think it is better to present the confirmed results regarding certain species and it i stoo early to propose a general concept on the figure.

Reviewer 2 Report

The text is very well written, and the references are exhaustive and seem up to date. References are not always pertinent cause sometimes they refer to other species and that is not clearly stated in the text. The title is clear and punchy, and rises a pertinent question, but in my opinion the article content does not respond adequately to the question.

I think the article is overall not well organized, the content mixes foal information with adult horse information, in certain sections it becomes merely a compilation of article briefs, which has a tendency to lose the reader without clearly drawing a conclusion. There has been in recent years multiple review articles about probiotic use in horses, and I am not sure a review specifically about probiotics in foals is something specially needed, unless you treat the subject clearly independently than adults, and you defend why the subject deserves the attention of the public (i.e. microbiota of foals different than adults, or other physiological differences between foals and adults that makes that different probiotic could apply or that response could be different than adults).

Authors rise the concern of antimicrobial resistance, which is interesting it itself, but references are mostly from other species and no information specifically in horses/foals is found, so their conclusions have not a solid basis despite the original question is pertinent.

Definition of what authors consider "a foal" is not stated in the article and definitely could help to determine the pertinence or not of certain sections, pathologies (i.e. <1 month foals, <6 mo foals, < 1 year?)

Specific comments:

Title: too ambitious for the content and the solidity of the conclusions obtained, specifically on the subject of the "risky gamble".

Abstract: 

Lines 15-18: Sentence too long and not sure if grammatically correct. Consider reformulating.

Line 21: pros and cons ...of what? i.e. its administration, its use... 

Introduction:

lines 39-41 : Not sure of the pertinence of this sentence in this section.

Section 6: Equine probiotic strains

This section is overall non-foal oriented, so not sure about the pertinence following the title of the article. It could be shortened to show the main conclusions about the probiotic strains used/proposed for use in the equine species.

Lines 216-218: and probably also because they are the main sites of microbial activity and fermentation  in the equine GI tract.

Line 236: most commonly used...in other animal species

Line 241: The foal core microbiota has not been clearly described as such, to my knowledge. The references you mention do not speak about foals. 

Section 7: Probiotics in foals

Page 11. Line 41; Groups "of" fecal samples

Section 8: Safety of probiotics

Page 13,Lines 129-131: I do not agree with this sentence. Probiotics are not an alternative to antimicrobials and the do not pretend to be. On the other hand probiotics may help to compensate/minimize the adverse (non selective) effects of antimicrobials on Gi tract microbiota.

Author Response

REVIEWER 2

Comments and Suggestions for Authors

The text is very well written, and the references are exhaustive and seem up to date. References are not always pertinent cause sometimes they refer to other species and that is not clearly stated in the text. The title is clear and punchy, and rises a pertinent question, but in my opinion the article content does not respond adequately to the question.

The title has been changed to be less dramatic.

I think the article is overall not well organized, the content mixes foal information with adult horse information, in certain sections it becomes merely a compilation of article briefs, which has a tendency to lose the reader without clearly drawing a conclusion. There has been in recent years multiple review articles about probiotic use in horses, and I am not sure a review specifically about probiotics in foals is something specially needed, unless you treat the subject clearly independently than adults, and you defend why the subject deserves the attention of the public (i.e. microbiota of foals different than adults, or other physiological differences between foals and adults that makes that different probiotic could apply or that response could be different than adults).

The article has been reorganized, the order of sections has been changed according to Reviewer 1’s suggestion. We hope the differences between adult horses and foals and the need for special attention for foals are more clearly empasized now. For example we explain the target of equine probiotics is generally presumed to be the hindgut, but foal diarrhoea often results from processes in the small intestine (lines 289-300)

Authors rise the concern of antimicrobial resistance, which is interesting it itself, but references are mostly from other species and no information specifically in horses/foals is found, so their conclusions have not a solid basis despite the original question is pertinent.

Unfortunately, the literature regarding horses and especially foals in this field is limited, that is why we used additional references regarding other species. The issue of antimicrobial resistence can be extrapolated to horses, because at very least, regardless of the effect in horses themselves, excretion of antimicrobial-resistant strains into the environment is a concern in any species in which such strains are used.

Definition of what authors consider "a foal" is not stated in the article and definitely could help to determine the pertinence or not of certain sections, pathologies (i.e. <1 month foals, <6 mo foals, < 1 year?)

In regards to GIT microbiota, the age of foal is essential and one definition cannot be stated. The GIT microbiota develops and changes with age (as indicated in the section „Singularity of the Developing Foal Microbiota”). That is why we did not define “a foal” by age in general, but we indicated the age (1 month, less than one month, 15 days etc..) of foals when discussing certain points in the article. In response to this concern, we also clarified the age of foals included in studies in the table 2 (in section 7).

Specific comments:

Title: too ambitious for the content and the solidity of the conclusions obtained, specifically on the subject of the "risky gamble".

Has been changed to less dramatic: “Gut microbiota manipulation in foals - naturopathic diarrhea management, or unsubstantiated folly?”

Abstract: 

Lines 15-18: Sentence too long and not sure if grammatically correct. Consider reformulating.

In our opinion this sentence clearly reflects the issue and is grammatically correct. We can reword it if the Reviewer insists.  

Line 21: pros and cons ...of what? i.e. its administration, its use... 

The use. It has been clarified.

Introduction:

lines 39-41 : Not sure of the pertinence of this sentence in this section.

We would like to mention heat diarrhoea at the beginning as it is very common. That is why we left this sentence, however, it can be deleted if the Reviewer insists.

Section 6: Equine probiotic strains

This section is overall non-foal oriented, so not sure about the pertinence following the title of the article. It could be shortened to show the main conclusions about the probiotic strains used/proposed for use in the equine species.

This section was designed as a general background for the next, foal oriented section. We have reorganized this section, deleted some information and replace them by more details regarding foals.

Lines 216-218: and probably also because they are the main sites of microbial activity and fermentation  in the equine GI tract.

This information has been added

Line 236: most commonly used...in other animal species

Certainly, the genera listed in this sentence are the most commonly used in other animal species (as written later in this section- lines 329-331). In this sentence we would like to emphasize that what is used most commonly is not necessarily suitable for horses.

Line 241: The foal core microbiota has not been clearly described as such, to my knowledge. The references you mention do not speak about foals. 

We absolutely agree that the knowledge regarding the foal core microbiota is limited, as well as the knowledge regarding probiotic use. That is why we have decided to write this review based on the current literature. The references cited here contain the short section „Current Evidence for use of Probiotic in Foals” and limited information regarding foals can be found inside, that is why we cited this article in this place. We have cited more references (2, 5, 14, 15, 27) in the following sentence.

Section 7: Probiotics in foals

Page 11. Line 41; Groups "of" fecal samples

Has been changed

Section 8: Safety of probiotics

Page 13,Lines 129-131: I do not agree with this sentence. Probiotics are not an alternative to antimicrobials and the do not pretend to be. On the other hand probiotics may help to compensate/minimize the adverse (non selective) effects of antimicrobials on Gi tract microbiota.

We do agree that probiotcs are not an alternative to antimicrobials and sorry for this misunderstanding. Just in respect to the resistance, the goal is to avoid the overuse and prophylactic use of antimicrobials.

The sentences have been reworded: ”One of the drivers of interest in microbiota supplementation is a desire to reduce antimicrobial use in order to stem the spread of bacterial antimicrobial resistance. For example, probiotic GIT fortification is an attractive alternative to traditional prophylactic 

Reviewer 3 Report

The review article “Gut microbiota manipulation in foals – naturopathic diarrhea management, or unsubstantiated folly?” aimed to review the literature regarding use, efficacy, safety and challenges of probiotics, prebiotics and fecal microbial transplants primarily in foals, but also in comparison to adult horses. This is a MUCH-needed review, thank you to the authors! I really enjoyed reading this.

This manuscript was written with excellent English language and minor spelling and typographical errors (punctuation, spacing, font size, section numbering, italics, bold, etc.).

The content of the manuscript is thorough aside from a few suggested studies to add to Table 2 and body of text (it was unclear what the literature search criteria was for this review). Overall, nicely done!

Here are a few select specific recommendations for minor revisions:

Line 85, Page 2: substitute “sex” for “gender”

Lines 32 and 51, Page 12; Line 61, Page 13 (and in other locations throughout the manuscript): Because “neonate” is being used as an adjective, it should be “neonatal” in front of “Thoroughbred(s)” and “intestinal”.

Table 2:

Review the whole table for consistency in punctuation, spacing and spelling. The size of the “x” used for cfu’s is different between studies.

For the second study listed, remove “/day” from the dose column because this is described in the adjacent column. Also, for the second study listed, the outcomes as described don’t seem to answer the aim – is there a better way to present one or the other? Seems as though WE7 was a superior lactic acid bacteria in vitro and selected for further study. Then WE7 was inhibitory to E. coli, S. zoo and C. diff making it a potential preventive and/or treatment for equine enteric conditions.

For the sixth study listed, study limitation column, use “PG” for “placebo groups” to be consistent. Should this be split into 2 separate rows/studies since there are two references provided? 

For the eighth study listed, study limitation column, recommend saying the following for improved clarity “Used culture-based methods for pathogen screening.”

Add SCFAs to the list of defined abbreviations used in the table.

Other studies to consider adding to the table:

Changes in the fecal microbiota associated with broad-spectrum antimicrobial administration in hospitalized neonatal foals with probiotics supplementation. https://doi.org/10.3390/ani11082283

Effects of oral supplementation of probiotic strains of Lactobacillus rhamnosus and Enterococcus faecium on diarrhoea events of foals in their first weeks of life. https://doi.org/10.1111/jpn.12923

Efficacy of the probiotic strain Enterococcus faecalis CECT7121 in diarrhoea prevention in newborn foals. http://dx.doi.org/10.30972/vet.2711075

References:

There are several non-peer-reviewed references listed, including Vet Clinics of North America, AAEP Proceedings, websites, etc. I would highly encourage the authors to find peer-reviewed alternatives, especially references 3, 45 and 48 (to which I was taken to a nonexistent website). This will nicely improve the robustness and credibility of your review article.

Review references to ensure they match the Journal’s formatting – some are incomplete/incorrect (examples are 22, 56, etc.).

Line 264, Page 7: Ref 64 doesn’t seem to be suitable for this statement about a single study. Double check your references within the manuscript to ensure accuracy.

Author Response

The review article “Gut microbiota manipulation in foals – naturopathic diarrhea management, or unsubstantiated folly?” aimed to review the literature regarding use, efficacy, safety and challenges of probiotics, prebiotics and fecal microbial transplants primarily in foals, but also in comparison to adult horses. This is a MUCH-needed review, thank you to the authors! I really enjoyed reading this.

This manuscript was written with excellent English language and minor spelling and typographical errors (punctuation, spacing, font size, section numbering, italics, bold, etc.).

The content of the manuscript is thorough aside from a few suggested studies to add to Table 2 and body of text (it was unclear what the literature search criteria was for this review). Overall, nicely done!

 We would like to kindly thank the Reviewer for appreciating our work and this kind opinion.

All comments have been addressed. According to the reviewer’s suggestions, corrections have been made and additional references have been added. 

Here are a few select specific recommendations for minor revisions:

Line 85, Page 2: substitute “sex” for “gender”

Has been changed

Lines 32 and 51, Page 12; Line 61, Page 13 (and in other locations throughout the manuscript): Because “neonate” is being used as an adjective, it should be “neonatal” in front of “Thoroughbred(s)” and “intestinal”.

While we understand the reviewer’s point, "neonate" is actually being used intentionally as a noun modifier rather than as an adjective.  While “neonatal foal/Thoroughbred” would also be correct, we prefer the sound of "neonate" with respect to the rhythm of the sentences in which the phrases are used.  Since both are grammatically correct variations expressing the same meaning, we would prefer to leave “neonatal foal/Thoroughbred” as is. However, we agree to change “neonate intestinal microbiota” to “neonatal”, as we agree “neonatal” is more appropriate to describe the microbiota that pertains to the neonatal period.

Table 2:

Review the whole table for consistency in punctuation, spacing and spelling. The size of the “x” used for cfu’s is different between studies.

All above have been corrected. We hope it will not be reformatted when submitted.

For the second study listed, remove “/day” from the dose column because this is described in the adjacent column.

Has been removed

Also, for the second study listed, the outcomes as described don’t seem to answer the aim – is there a better way to present one or the other? Seems as though WE7 was a superior lactic acid bacteria in vitro and selected for further study. Then WE7 was inhibitory to E. coli, S. zoo and C. diff making it a potential preventive and/or treatment for equine enteric conditions.

Has been changed into:” In vitro testing, WE7 performed thebest with respect to the ability to grow in the presence of oxygen, aid and bile, The isolate was inhibitory against Salmonella and E. coli, moderately inhibitory against S. zooepidemicus and C. difficile and midly inhibitory against  C. perfringens. After demonstrating the ability to survive GIT passage, WE7 was thus selected for further study of its potential to prevent and treat enteric diseases in horses.”

For the sixth study listed, study limitation column, use “PG” for “placebo groups” to be consistent.

Has been changed

Should this be split into 2 separate rows/studies since there are two references provided? 

The same strains in the same doses have been used in these studies (refs 9 and 53), therefore they were described together.

For the eighth study listed, study limitation column, recommend saying the following for improved clarity “Used culture-based methods for pathogen screening.”

Has been changed

Add SCFAs to the list of defined abbreviations used in the table.

Has been added

Other studies to consider adding to the table:

Changes in the fecal microbiota associated with broad-spectrum antimicrobial administration in hospitalized neonatal foals with probiotics supplementation. https://doi.org/10.3390/ani11082283

While this study is very interesting and we thank the reviewer for bringing it to our attention, the study evaluates the effects of antimicrobials on the foal microbiota, not the effect of probiotics.  Had the study included a control group receiving antibiotics without probiotic supplementation and analyzed the effect of probiotics on the microbiota of foals undergoing systemic antimicrobial therapy, we would have eagerly included the results. We considered including the reference elsewhere in the article, but don’t feel comfortable citing it as evidence of anything given the small sample size and the confluence of factors for which there was no standardization or means of distinguishing the source of effects.  For example, only seven foals were studied in total.  All of the foals received antimicrobials and probiotics, but antimicrobial treatment duration varied, some foals received fecal microbial transplantation, and no foals served as a control for anything. We believe this study is an interesting starting point and should certainly be followed-up for a better understanding of the impacts of antimicrobial administration on the foal intestinal microbiota.  However, the results of this initial study are beyond the scope of our article.

Effects of oral supplementation of probiotic strains of Lactobacillus rhamnosus and Enterococcus faecium on diarrhoea events of foals in their first weeks of life. https://doi.org/10.1111/jpn.12923

This is the same study already cited, just described in a different article.  However, the additional reference has been added.

Efficacy of the probiotic strain Enterococcus faecalis CECT7121 in diarrhoea prevention in newborn foals. http://dx.doi.org/10.30972/vet.2711075

Has been added to the table.  Note the authors of this study specified foals received 1 x 10^10 cfu/ml, but not how many ml each foal received.  We also added mention of this study in the discussion following Table 2.

References:

There are several non-peer-reviewed references listed, including Vet Clinics of North America, AAEP Proceedings, websites, etc. I would highly encourage the authors to find peer-reviewed alternatives, especially references 3, 45 and 48 (to which I was taken to a nonexistent website). This will nicely improve the robustness and credibility of your review article.

Ref. 3 has been changed into AAEP Proceedings recommended by ivis.org. Ref. 45 (currently 41) has been replaced. Ref 44 (previosly 48) is an European regulation, so there is no peer reviewed alternative, the link has been changed directly inot this document

Review references to ensure they match the Journal’s formatting – some are incomplete/incorrect (examples are 22, 56, etc.).

Sorry for these mistakes. Have been corrected.

Line 264, Page 7: Ref 64 doesn’t seem to be suitable for this statement about a single study. Double check your references within the manuscript to ensure accuracy.

Sorry for this mistake, ref 64 (currently 60) has been moved into proper place